# Fluorescein Derivative Immobilized Optical Hydrogels: Fabrication and Its Application for Detection of H_2_O_2_

**DOI:** 10.3390/polym14153005

**Published:** 2022-07-26

**Authors:** Zixiang Qu, Demeng Zhang, Chuane Wang, Sheng Tian, Yunlong Deng, Dawei Qin, Hongdong Duan

**Affiliations:** 1School of Chemistry and Chemical Engineering, Qilu University of Technology (Shandong Academy of Sciences), Jinan 250353, China; qzxorc@163.com (Z.Q.); 15563222524@163.com (C.W.); daweiqin@163.com (D.Q.); 2State Key Laboratory of Bioactive Seaweed Substances, Qingdao Bright Moon Seaweed Group Co., Ltd., Qingdao 266400, China; zdm@bmsg.com (D.Z.); yunlong.deng@bmsg.com (Y.D.); 3Chemical Technology Academy of Shandong, Qingdao University of Science & Technology, Jinan 250013, China; sdscce@163.com

**Keywords:** hydrogen peroxide, hydrogel sensor, naked-eye detection

## Abstract

A novel fluorescein-based probe **FLA-Boe** was developed for detecting H_2_O_2_. Modified by 2-Bromomethylphenylboronic acid pinacol ester, **FLA-Boe** is a Fluorescein derivative with eminent photostability and remarkable H_2_O_2_ sensitivity and selectivity. **FLA-Boe** was utilized to synthesize hydrogel sensors in the manner of guest–host interaction by taking advantage of its aforementioned features. The hydrogel sensor can be used to detect H_2_O_2_ effectively in both flowing and static water environments with satisfactory performance. It is expected that this application may open a new page to develop a neoteric fluorescent property analysis method aiming at H_2_O_2_ detection.

## 1. Introduction

Hydrogen peroxide (H_2_O_2_), a typical reactive oxygen species (ROS), functions as a key throughout cell growth, proliferation, host defense, and signal transmission pathways under a physiological environment [1,2]. However, the chemical and pharmaceutical industries inevitably use hydrogen peroxide as an oxidizer for related production activities, resulting in the discharge of excessive hydrogen peroxide in wastewater, which severely impacts people’s living environment and the water treatment system [3]. Excessive H_2_O_2_ in the human body triggers the pathogenesis of many disorders, such as inflammation, Alzheimer’s disease, cardiovascular disease and, more seriously, cancer [4]. Consequently, it is vital to monitor H_2_O_2_ in the inner condition and outer environment of the human body [5]. Undoubtedly, the development of H_2_O_2_ fluorescent sensors has been arousing people’s interest. Thus, the design and fabrication of sensing systems with H_2_O_2_ responsive characteristics are key for human health and environmental protection.

Macroscopic supramolecular hydrogels are excellent candidate materials for intelligent sensing activity. In comparison with naturally existing hydrogels, such as polysaccharides and proteins, assorted synthetic hydrogels have been applied as smart materials on account of their functionalization during synthesis. Furthermore, most hydrogels are economical and environmentally friendly with transparent and stretchable features [6,7]. Nowadays, composite materials based on macroscopic supramolecular hydrogels have been investigated for optical detection. Using hydrogel sensors to detect test objects in environmental samples is a handy, easily operated, low-cost, analytical method with remarkable sensitivity [8]. Through research, it is possible to conduct naked-eye observation and direct analysis without spectroscopic devices and apparatus. Hitherto, diverse synthetic strategies using hydrogel sensors have been extensively explored and studied, accompanied by their functionalization procedures, performance control, and potential applications [9,10,11,12,13]. According to experience, device-free sensor systems aiming at colorimetric and fluorometric detection by the naked eye are of the most extensive and applicable utility ad hoc; the aforementioned hydrogel sensors are easily manipulated, as well as featuring eminent sensitivity and high signal-to-noise ratio [14,15]. Furthermore, these materials can be easily and cheaply prepared, and are highly adaptive to various conditions. As mentioned above, cyclodextrin moieties can be incorporated into pre-existing polymeric materials via grafting reactions. Interpenetrated networks can also be used to develop new selective and synergistic sorption capacities for specific purposes, such as the fabrication of a hydrogel sensor. Surmounting the shortcomings existing among traditional fluorescence detection methods, such as monitoring limited, complicated detection processes and man-made mistakes requires the application of hydrogel immobilized sensors with response signals that are adaptive to the living environment and that are able to simplify the detection operation and acquire more convincing and urgently needed detection results [16,17,18,19]. Thus, the development of hydrogel sensors for detecting H_2_O_2_ with dual colorimetric and fluorescence signals is of profound significance. Crosslinking and grafting are applied to prepare covalently-attached cyclodextrins [20,21]. The application of a supramolecular sensing system has a promising application prospect [22]. With regard to complexity, functionality and capabilities, the evolution of cyclodextrin polymers matches the modern requirements of macromolecular materials.

In this study, fluorescein-based derivatives are selected as the fluorophores by structural modifications and the phenylboronic acid ester acts as the recognition site for H_2_O_2_. The collected results underpinned the development of a responsive fluorescence probe, in an aqueous medium, that has the potential for discovering hydrogen peroxide in the inner condition and outer environment of the human body. Hence a novel fluorescence probe **FLA-Boe** was devised and synthesized, which exhibited ideal properties for H_2_O_2_ detection. By utilizing this probe, a transparent optical hydrogel sensor with a fluorescence feature for H_2_O_2_ sensing was prepared and synthesized; the result remained effective based on this hydrogel sensor after a series of experiments were conducted. After being exposed to H_2_O_2_, the hydrogels were colored purplish red; the intensity was associated with H_2_O_2_ concentration. Thus, the hydrogel sensor features selective and integrated detection and monitoring of H_2_O_2_ within one system. This effective approach to developing a novel macroscopic hydrogel sensor provides a convenient and intelligent sensing system that is adaptive to complex aqueous environments.

## 2. Materials and Methods

### 2.1. Materials and Instrumentation

Fluorescein and 2-Bromomethylphenylboronic acid pinacol ester were purchased from Aladdin Co., Ltd. (Jinan, China) All chemical reagents reached analytical grade or obtained the highest purity availability, and were free of further purification. The preparation in terms of metal ions solutions was conducted in deionized water. ^1^H and ^13^C NMR spectra were recorded on a Bruker 400 MHz spectrometer (^1^H 400 MHz; ^13^C 100 MHz, Jinan, China) by using DMSO-*d*_6_ as solvents. FT-IR spectra were obtained on a Thermo fisher Nicolet 6700 FT-IR spectrophotometer (Jinan, China). Q-TOF LC/MS mass spectrometry acted as the recorder for high-resolution mass spectra (HRMS). Shimadzu UV-2600 spectrophotometer was utilized to perform UV–Vis absorption spectra (Jinan, China). Fluorescence spectra were measured on Hitachi F-4600 (Jinan, China) which functions as the gauge of fluorescent intensity.

### 2.2. Preparation of Samples and Test Solution

Solutions of 10^−3^ M of the test object—NaCl, Na_2_CO_3_, NaNO_3_, NaNO_2_, Na_2_S NaOCl, glutathione (GSH), Glutamic acid (Glu), Histidine (His), Levodopa (L-DOPA), Lactate (Lac), Fructose (Fru), Glucose (Glc), Urea, Alanine (Ala), Choline oxidase (ChOx), glucose oxidase (GOD)—were prepared in deionized water. The stock solution (10^−3^ M) was prepared by dissolving **FLA-Boe** in DMSO. The DMSO/H_2_O mixed solution diluted the **FLA-Boe** solution to acquire the analytical solution (DMSO/H_2_O = 1/1, *v*/*v*). The pH scope of solutions was adjusted by 4 M of hydrochloric acid aqueous solution and 4 M of sodium hydroxide aqueous solution.

### 2.3. Synthesis of FLA-Boe

As shown in Figure 1, Fluorescein (3.3230 g, 1.00 mmol) and hydrazine monohydrate (50%, 3 mL) was dissolved in 100 mL of methanol, after which the mixture was heated to reflux for 4 h. Then the solvent was evaporated, and the resultant residue was subjected to column chromatography (dichloromethane/ethanol = 20/1, *v*/*v*) to obtain the intermediate *N*-(Fluorescein) lactam (2.9611 g, 0.86 mmol). *N*-(Fluorescein)lactam (3.4634 g, 1.00 mmol) and acetic acid (0.5 mL) were dissolved in 150 mL of acetone and the mixture was heated to 50 °C and stirred for 5 h. The mixture was filtered and the filter cake was washed with acetone and then purified by column chromatography (dichloromethane: ethanol = 30/1, *v*/*v*) to obtain the *N*-(Fluorescein) lactam-*N*′-methylethylidene (2.7814 g, 0.75 mmol). *N*-(Fluorescein) lactam-*N*′-methylethylidene (3.8640 g, 1.00 mmol) was added into a flask containing the mixture of 4-Bromomethylphenylboronic acid pinacol ester (2.3760 g, 0.8 mmol), K_2_CO_3_ (3.0 g), and 20 mL of DMF at 40 °C for 6 h, and the mixture was poured into H_2_O and extracted with ethyl acetate. The organic phase was separated and dried by MgSO_4_, meanwhile, the solvent was removed by vacuum distillation. **FLA-Boe** was obtained as the white solid after being purified by column chromatography with dichloromethane/ethanol (40/1, *v*/*v*) as eluent (2.5295 g, yield = 42%). ^1^H NMR (400 MHz, DMSO-*d*_6_, ppm): δ 9.86 (d, *J* = 15.1 Hz, 1H), 7.86–7.79 (m, 1H), 7.70 (d, *J* = 7.9 Hz, 2H), 7.58–7.52 (m, 2H), 7.46 (d, *J* = 7.8 Hz, 2H), 7.05 (p, *J* = 4.4 Hz, 1H), 6.85 (t, *J* = 3.7 Hz, 1H), 6.69 (dd, *J* = 8.7, 2.4 Hz, 1H), 6.61–6.57 (m, 1H), 6.57 (s, 1H), 6.53–6.50 (m, 1H), 6.48 (dd, *J* = 8.0, 5.9 Hz, 1H), 5.17 (s, 2H), 1.85 (s, 3H), 1.76 (s, 3H), 1.29 (s, 12H). ^13^C NMR (101 MHz, DMSO-*d*_6_, ppm): δ 174.00, 160.47, 159.35, 158.81, 152.59, 152.49, 140.60, 135.09, 133.34, 130.07, 129.10, 127.35, 123.26, 112.32, 102.66, 84.17, 69.73, 65.24, 25.41, 25.15, 21.73. FT-IR: (KBr, cm^–1^) ν 2981.31 (O–H) 1696.71 (C=O) 1614.28 (C=N) 1362.12 (C=C) 1176.55 (C–O), HRMS (ESI): C_36_H_35_BN_2_O_6_ calculated for: 602.2588; Found 603.2827 for [M+H]^+^.

### 2.4. Synthesis Process of Hydrogel Sensor

The hydrogel sensors were prepared and synthesized by guest–host interaction as shown in Figure 1. At first, Acrylamide (AAm), β-cyclodextrin (β-CD) and *N*,*N*′-methylene bisacrylamide (MBA) (0.3% of monomer content) were mingled together and dissolved in DMSO solution. Then the mixture was stirred under a vacuum at 5 °C until a homogeneous solution was formed. In the next step, 100 μL of TEMED and initiator (APS, 2% of monomer content) were put into the mixture. The mixture was stirred constantly until a homogeneous solution was witnessed during the reaction, which was subsequently poured into a cylindrical mold and maintained at 50 °C for 12 h. In the following step, the hydrogel was extracted and collected from the mold and then rinsed with DMSO for 36 h. AAm-co-β-CD hydrogel was steeped in 10^−2^ mol/L of **FLA-Boe** DMSO solution for 10 h before being dialyzed in H_2_O for 2 h to fabricate the hydrogel sensor in accordance with the host–guest interaction.

## 3. Results

### 3.1. Optical Properties of FLA-Boe

The optical properties of **FLA-Boe** were performed and analyzed with the help of UV–Vis and fluorescence spectroscopy in 1.0 × 10^−3^ mol/L HEPES buffer (DMSO/H_2_O = 1/1, *v*/*v*, pH = 7.4) through various test objects. As shown in Figure 2a, the absorption band of the solution of **FLA-Boe** (10^−4^ mol/L, DMSO/H_2_O, 1/1, *v*/*v*) notably reached 300–700 nm after investigation, and the addition of H_2_O_2_ performed an obvious absorption band reaching 417 nm. The result demonstrated that the **FLA-Boe** had a selective response to H_2_O_2_. Nevertheless, no other clear absorption band appeared after adding other test objects, thus the phenomenon proved the prominent selectivity of **FLA-Boe** toward H_2_O_2_.

The fluorescence emission property of **FLA-Boe** toward different test objects (H_2_O_2_, NaCl, Na_2_CO_3_, NaNO_3_, NaNO_2_, Na_2_S, NaClO, glutathione (GSH), glutamic acid (Glu), histidine (His), levodopa (L-DOPA), lactate (Lac), fructose (Fru), glucose (Glc), urea, alanine (Ala), choline oxidase (ChOx), and glucose oxidase (GOD)) was analyzed and investigated in DMSO and H_2_O solution (1/1, *v*/*v*), as shown in Figure 2b. From the standard solution of **FLA-Boe** in DMSO and H_2_O (1/1, *v*/*v*), no distinct fluorescence emission was performed. However, as long as H_2_O_2_ was added, significant fluorescence emission was displayed, whose emission level was maximized at 526 nm, which could indirectly affect the π-electron push–pull function and resulted in the intramolecular charge transfer (ICT). Additionally, the color change of the solution from transparent to red clearly appeared (Figure 2c). For experiment comparison, other metal ions were added in the solution, and the fluorescence spectra remained unchanged. Hence the result demonstrated eminent fluorescent selectivity of **FLA-Boe** towards H_2_O_2_.

### 3.2. Sensitivity and Selectivity

As shown in Figure 3a, the fluorescence titration experiments were conducted. **FLA-Boe** solution was incubated by changing the amounts of H_2_O_2_ in pH 7.4 HEPES buffer solution. On the basis of the aforementioned investigation, the emission of **FLA-Boe** is dominated by the intense excimer emission (DMSO/H_2_O = 1/1, *v*/*v*). When the probe was excited at 417 nm, the fluorescence intensity evidently increased at 526 nm after the addition of H_2_O_2_. **FLA-Boe** rapidly responded to H_2_O_2_ and the fluorescence intensity was distinctly boosted, which could be caused by the intramolecular charge transfer (ICT) process. Phenylboronic ester in **FLA-Boe** was cleaved through an oxidative reaction when contacting with H_2_O_2_, which resulted in the enhancement of fluorescence intensity. As shown in Figure 3b, the detection limit (DL) was investigated and obtained from the data, reaching 0.22 μM.

### 3.3. Competition Experiments

To further check the practical applicability of receptor **FLA-Boe** as the selective fluorescent sensor for H_2_O_2_, competitive experiments were carried out with 2.0 equiv of H_2_O_2_ in the presence of other test objects in an aqueous solution, as shown in Figure 4. It was interesting to note that only H_2_O_2_ rendered significant fluorescence turn-on responses among all the tested objects, whereas all the coexistent test objects had no noteworthy variance with regard to fluorescence intensity. These investigated results clearly demonstrated that the sensor **FLA-Boe** for detecting H_2_O_2_ had no interference and could be a good sensor for H_2_O_2_ detection in aqueous media.

### 3.4. Detection Mechanism of FLA-Boe

The presupposition of the **FLA-Boe** probe design aims to block the possible ICT process by protecting the hydroxyl group with boronate ester. Hence, Fluorescein was selected as the fluorophore and phenylboronic acid pinacol ester acted as the receptor unit. The inferred detection mechanism of the probe **FLA-Boe** was recommended in Figure 2. Due to the protection mechanism of the boronate ester on the hydroxyl group, the ICT process of **FLA-Boe** was hindered, which demonstrates no fluorescence emission of **FLA-Boe** was triggered. While **FLA-Boe** was exposed to H_2_O_2_, the decomposition of the boronate ester group was taken and the hydroxyl group was released; then the ICT process of Fluorescein was recovered, through which the strong fluorescence emission from Fluorescein was inevitably triggered. The FT-IR and HRMS tests were investigated to deduce the rational mechanism. In order to verify the response mechanism in detail, the generation of Compound **2**—the product of **FLA-Boe** and H_2_O_2_ reaction—was monitored by means of HRMS. As shown in Figure 5a, FT-IR spectrum results confirmed the reasonableness of the aforementioned response mechanism hypothesis; after the addition of the H_2_O_2_, a notable hydroxyl peak appeared at around 3250 cm^−1^. Moreover, the HRMS result proved that Compound **2** was the product of the **FLA-Boe** and H_2_O_2_ reaction (Figure 5b). The inference concerning the fluorescence response mechanism of **FLA-Boe** to H_2_O_2_ was preliminarily confirmed from these results of the experiments.

### 3.5. The pH Effect

The influence and phenomenon of the probe **FLA-Boe** to H_2_O_2_ were analyzed in different pH conditions. As shown in Figure 6, within the scale of pH 3.0–4.0, the probe failed to respond notably to H_2_O_2_. While **FLA-Boe** was exposed to H_2_O_2_ within the scope of pH 5.0–8.0, the decomposition of the boronate ester group was taken, then the ICT process of Fluorescein was recovered, through which the strong fluorescence emission from Fluorescein was triggered. As a result, the **FLA-Boe** could respond efficiently to H_2_O_2_ in the pH range of 5.0–8.0. The fluorescence intensity reached 581 nm and the response of the probe to H_2_O_2_ gradually receded after the pH value was improved to 8.0–9.0. Thus, the phenomenon illustrated that the probe **FLA-Boe** had eminent fluorescent responses to H_2_O_2_ from weakly acidic to weakly alkaline conditions.

### 3.6. The Fabrication of Hydrogel Sensor

The work noted that the formation of the β-CD-**FLA-Boe** inclusion complex was distinctly performed by ^1^H NMR; meanwhile, the inclusion procedure was demonstrated preliminarily. The ^1^H NMR spectrum of β-CD during the appearance and disappearance of **FLA-Boe** was shown in Figure 7. The integral signal strengths in the ^1^H NMR spectrum were investigated. The inclusion of β-CD with **FLA-Boe** exhibited small changes, and a conspicuous influence was shown on the chemical shifts of protons of β-CD-**FLA-Boe**, which varied notably (Table 1), revealing that these protons primarily contributed to the inclusion with **FLA-Boe**. More supporting evidence regarding the inclusion of **FLA-Boe** in the central cavity of β-CD was acquired from ^1^H NMR studies. In comparisons of the free β-CD and **FLA-Boe**, the chemical shifts of the inclusion complexes were investigated, demonstrating that **FLA-Boe** had mutually interacted with β-CD.

### 3.7. Detection Performance of the Hydrogel Sensor

The hydrogel sensor responded conspicuously to H_2_O_2_ with prominent selectivity and sensitivity due to the role of ICT. To ascertain the capability and effectiveness of the hydrogel sensor, hydrogel detection limits were adjusted and confirmed among varied concentrations of H_2_O_2_ solutions, from 10^−4^ to 10^−7^ mol/L. By observation, the color of the hydrogel sensor could still be notably changed when H_2_O_2_ concentration reached 10^−6^ mol/L. The color of the hydrogel sensor changed when H_2_O_2_ concentration was 10^–4^, 10^−5^, and 10^−6^ mol/L for 50 s, 4 min, and 20 min, respectively. In Figure 8, the hydrogel sensor performed visible red light in the aqueous solution environment, which was the same as **FLA-Boe**’s; under 365 nm UV light, the color of the hydrogel sensor in the aqueous solution also conspicuously changed.

A comparison of the results indicates that the Fluorescein derivative immobilized system in this work exhibits better ability for H_2_O_2_ detection compared to the other previously reported systems (Table 2). In addition, many of the reported sensors involve poorer reuse performance. On the other hand, the present sensing system is cost-effective, selective, exhibits simple naked-eye sensing, and is suggested to be utilized to detect H_2_O_2_ in a real water environment.

### 3.8. Universality of the Hydrogel Sensor

Inspired by the remarkable function of the hydrogel sensor in solution for H_2_O_2_, the response of the hydrogel sensor for H_2_O_2_ in flowing and static water environments was further investigated. The hydrogel sensor was colored from transparent to light aubergine (about 6 min) after the sensor was exposed to the flowing H_2_O_2_ solution (10^−4^ mol/L). The color changed to sharp aubergine after 10 min. In a large container insulated from the atmosphere, the hydrogel sensor was exposed to a static H_2_O_2_ solution. An interesting fact was that when the gel was steeped in 10^−7^ mol/L H_2_O_2_ solution for 3 h, the hydrogel sensor changed color. During the experiment for detection performance, the hydrogel sensor had no response when contacting a small quantity of H_2_O_2_ solution whose concentration was no more than 10^−7^ mol/L. In accordance with the experimental phenomenon, it was summarized that a cumulative effect may appear in the hydrogel sensor that could result in the reduction of the detection limit to some extent. Thus, analyzing the sensing performance and process of the hydrogel sensor can open a new avenue for developing significant, long continuous, and fast-responding test strips for precisely detecting H_2_O_2_ by convenient naked-eye observation, which is of particular importance in the immediate monitoring of H_2_O_2_ in both natural aqueous environments and liquid environments around chemical industrial regions. Meanwhile, observing color changes by the naked eye simplifies and speeds up the detection of H_2_O_2_. Therefore, this technological proposal will be a reliable method and a suitable selection for the detection of H_2_O_2_ in aqueous environments.

## 4. Conclusions

In summary, based on the H_2_O_2_ probe **FLA-Boe**, we developed an easily operated and commonly applicable strategy to synthesize a H_2_O_2_ responsive hydrogel sensor with on–off switching and color-changing fluorescence features. Hydrogel sensors were utilized smoothly for the determination of trace amounts of H_2_O_2_ in water samples, which indicates that hydrogel sensors are available for use by common people and scientists depending on practical requirements. Additionally, this method has potential applications in establishing novel monitoring and detection mechanisms and systems that cover detection systems for flowing and static water environments, etc. It is expected that this method may provide diverse and novel strategies to exploit smarter systems that include functions, such as synergistic visual detection and efficient sensing.

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
