# Peer review of "Fluorescein Derivative Immobilized Optical Hydrogels: Fabrication and Its Application for Detection of H2O2"

_polymers, 2022, doi:10.3390/polym14153005_

Round 1

Reviewer 1 Report

The manuscript with the title “Fluorescein Derivative Immobilized Optical Hydrogels: Fabrication and its Application for Detection of H2O2” seem to be very interesting, but the manuscript cannot be published in the present form due to the following issues:

1.      For the clear observation the response curves within the Figure 2 (a –b) in absence of H2O2 should be zoom-in with extra figures

2.      Is the experiment performed single time. Error bars are absent in Figure 4 and figure 6

3.      The reasoning for the change in the response curve of FLA-Boe in presence of H2O2 within Fig 6 between 5 to 8 pH is incomplete

4.       Author should involve the novelty of the system which should be added before the conclusion part

Author Response

Comments and Suggestions for Authors

The manuscript with the title “Fluorescein Derivative Immobilized Optical Hydrogels: Fabrication and its Application for Detection of H2O2” seem to be very interesting, but the manuscript cannot be published in the present form due to the following issues:

  1. For the clear observation the response curves within the Figure 2 (a –b) in absence of H2O2 should be zoom-in with extra figures

Response: Thank you for the suggestion and we have rearranged the layout of Figure 2 (a –b).

  1. Is the experiment performed single time. Error bars are absent in Figure 4 and figure 6

Response: Thank you for your valuable advice and we have included error bars in Figure 4 and Figure 6.

  1. The reasoning for the change in the response curve of FLA-Boe in presence of H2O2 within Fig 6 between 5 to 8 pH is incomplete

Response: Thank you for the suggestion and we have added detailed explanations in section 3.5.

  1. Author should involve the novelty of the system which should be added before the conclusion part

Response: Thank you for the suggestion and we have described the novelty of the system in section 3.8.

Reviewer 2 Report

(1)  A previously published article (Kim et al. Chem Commun (Camb). 2014 Aug 25;50(66):9353-6. doi: 10.1039/c4cc02943g.) reported that boronate-ester bound to fluorescent molecules can be used to detect an oxidant. Although the fluorescent molecules are different, the authors should refer and cite this article because the basic concept is similar.    (2)  What are the ideas of the authors on the responsiveness of the method with oxidants other than H2O2? if it is not specific for H2O2 and detects oxidants in general, it cannot fully claim this method as an H2O2 detection method. At best, the concept of the article itself needs to be revised to say that it can detect H2O2 if H2O2, not other oxidants, is there.    (3) As a minor point, please state the relationship between the general material name (e.g. IUPAC name) and the abbreviation when the abbreviation first appears. For example, AAm, β-CD, MBA, etc.

Author Response

Comments and Suggestions for Authors

(1)  A previously published article (Kim et al. Chem Commun (Camb). 2014 Aug 25;50(66):9353-6. doi: 10.1039/c4cc02943g.) reported that boronate-ester bound to fluorescent molecules can be used to detect an oxidant. Although the fluorescent molecules are different, the authors should refer and cite this article because the basic concept is similar.   

Response: Thank you for the suggestion and we have referred and cited this article.

(2)  What are the ideas of the authors on the responsiveness of the method with oxidants other than H2O2? if it is not specific for H2O2 and detects oxidants in general, it cannot fully claim this method as an H2O2 detection method. At best, the concept of the article itself needs to be revised to say that it can detect H2O2 if H2O2, not other oxidants, is there.   

Response: We are not solely to investigate using boronate-containing fluorogenic compounds for detecting ROS and RNS, that's not our only line of research. Arylboronates react with ONOO- to yield corresponding hydroxyl derivatives much faster than with H2O2, but our investigations are not focus on the distinguish or identify between ROS and RNS, we are hoping to figure out effective approach to develop a novel macroscopic hydrogel sensor provides a convenient and intelligent sensing system that is adaptive to complex aqueous environments. The similar oxidants to H2O2 that widely present in the aquatic environment, such as superoxide anion, singlet oxygen, hypochlorite and hydroxyl radical have weak interference effect in detection of H2O2 by arylboronates.

(3) As a minor point, please state the relationship between the general material name (e.g. IUPAC name) and the abbreviation when the abbreviation first appears. For example, AAm, β-CD, MBA, etc.

Response: Thank you for your valuable advice and we have added the general material name when the abbreviation first appears.

Round 2

Reviewer 1 Report

As authors have addressed all the comments raised in their manuscript. Therefore  I recommend it for acceptance in Polymers.

Reviewer 2 Report

I have no further questions. The paper is of a sufficient level to be published and I recommend its acceptance.

This manuscript is a resubmission of an earlier submission. The following is a list of the peer review reports and author responses from that submission.

Round 1

Reviewer 1 Report

This paper describes a method of detecting H2O2 using a fluorescent composite material (FLA-Boe hydrogel). The experiments are well done and there are no serious flaws in the paper. However, I did not understand the purpose of this paper. In my opinion, the authors have to consider carefully again on the following points   (1) There have been many studies of fluorescent materials for detecting H2O2. Compared to them, FLA-Boe is not very sensitive, and its chemical structure is not very novel in terms of fluorecein-based arylboronate. What are the advantages of FLA-Boe compared to the existing reported H2O2 detection probes?   (2) The application of the FLA-Boe hydrogel sensor is unclear. How can this material be used to determine the "inner condition and outer environment of the human body" as described by the authors in the introduction? Also, what are the advantages of using this material over conventional materials?

Author Response

This paper describes a method of detecting H2O2 using a fluorescent composite material (FLA-Boe hydrogel). The experiments are well done and there are no serious flaws in the paper. However, I did not understand the purpose of this paper. In my opinion, the authors have to consider carefully again on the following points  

(1) There have been many studies of fluorescent materials for detecting H2O2. Compared to them, FLA-Boe is not very sensitive, and its chemical structure is not very novel in terms of fluorecein-based arylboronate. What are the advantages of FLA-Boe compared to the existing reported H2O2 detection probes?   

Response: Among the numbers H2O2 probes, the off-on sensors have better applicability, utilized these probes to synthesis hydrogel sensors, which can extend the application of H2O2 probes. As for the hydrogel sensor, the preparation method of host–guest interaction is more creative in probe selection and integration of different systems.

(2) The application of the FLA-Boe hydrogel sensor is unclear. How can this material be used to determine the "inner condition and outer environment of the human body" as described by the authors in the introduction? Also, what are the advantages of using this material over conventional materials?

Response: Hydrogel as a supramolecular affinitive material, it has great application potential in biological related fields. Meanwhile, H2O2 is an important Reactive Oxygen Species (ROS) in the inner condition and outer environment of the human body; this work is devoted to provide an idea for the application in detecting a variety of physiological activities related to H2O2. The advantages of using hydrogel over conventional materials are its’ excellent designability, relatively cheap price and easy to compound with other functional systems.

Reviewer 2 Report

This manuscript reports on the “Fluorescein derivative immobilized optical hydrogels: Fabrication and its application for detection of H2O2.” The content of the work is interesting since the hydrogel sensor can be used to detect H2O2 effectively in both flowing and static water environment with satisfactory performance. It is expected that this application may open a new page to develop a neoteric fluorescent property analysis method aiming at H2O2 detection. The manuscript can be published in the present form after fulfilling the minor issues:-

  1. Comparative detection of H2O2 through other hydrogels should be enlisted in tabular form.
  2. The author should raise are real issues points that have been solved with this kind of hydrogels?

Author Response

This manuscript reports on the “Fluorescein derivative immobilized optical hydrogels: Fabrication and its application for detection of H2O2.” The content of the work is interesting since the hydrogel sensor can be used to detect H2O2 effectively in both flowing and static water environment with satisfactory performance. It is expected that this application may open a new page to develop a neoteric fluorescent property analysis method aiming at H2O2 detection. The manuscript can be published in the present form after fulfilling the minor issues:

  1. Comparative detection of H2O2 through other hydrogels should be enlisted in tabular form.

Response: Thank you for the suggestion and we have added the Comparison table for hydrogel sensors systems in section 3.7.

  1. The author should raise are real issues points that have been solved with this kind of hydrogels?

Response: Hydrogel sensor mentioned in this study has satisfactory sensing performance, we believe it can dramatically improve the performance of application in H2O2 detection, and the synthesis method of hydrogel sensor could inspire the preparation of other supramolecular sensing systems.

Round 2

Reviewer 1 Report

I have no particular comment.